# Results and evaluation of the expansion of a model of comprehensive care for Chagas disease within the National Health System: The Bolivian Chagas network

Maria-Jesus Pinazo[1,4]*, Mirko Rojas-Cortez[2], Ruth Saravia[2], Wilson Garcia-Ruiloba[2], Carlos Ramos[2], Jimy-Jose Pinto Rocha[2], Lourdes Ortiz[2,3], Mario Castellon[2], Nilce Mendoza-Claure[2], Daniel Lozano[2], Faustino Torrico[2], Joaquim Gascon[1,4], on behalf of Chagas Platform and Chagas Healthcare Network working group[¶]

1 Barcelona Institute for Global Health (ISGlobal), Hospital Clínic- Universitat de Barcelona, Barcelona, Spain, 2 Fundación CEADES, Cochabamba, Bolivia, 3 Universidad Autónoma Juan Misael Saracho, Tarija, Bolivia, 4 Centro de Investigación Biotecnológica en Red de Enfermedades Infecciosas (CIBERINFEC)

¶ Membership of Chagas Platform and Chagas Healthcare Network working group is listed in the acknowledgments
* mariajesus.pinazo@isglobal.org

## Abstract

### Background

Most people with chronic Chagas disease do not receive specific care and therefore are undiagnosed and do not receive accurate treatment. This manuscript discusses and evaluates a collaborative strategy to improve access to healthcare for patients with Chagas in Bolivia, a country with the highest prevalence of Chagas in the world.

### Methods

With the aim of reinforcing the Chagas National Programme, the Bolivian Chagas Platform was born in 2009. The first stage of the project was to implement a vertical pilot program in order to introduce and consolidate a consensual protocol-based healthcare, working in seven centers (Chagas Platform Centers). From 2015 on the model was extended to 52 primary healthcare centers, through decentralized, horizontal scaling-up. To evaluate the strategy, we have used the WHO ExpandNet program.

### Results

The strategy has significantly increased the number of patients cared for, with 181,397 people at risk of having *T. cruzi* infection tested and 57,871 (31·9%) new diagnostics performed. In those with treatment criteria, 79·2% completed the treatment. The program has also trained a significant number of health personnel through the specific Chagas guidelines (67% of healthcare workers in the intervention area).

**Data Availability Statement:** All relevant data are within the manuscript and its Supporting Information files. Anonymized patients data are available at Chagas Platform server (Fundación CEADES, Bolivia), and databases are shared between CEADES Foundation host with biostatistics department at ISGlobal. Data underlying this article will be shared on reasonable request to Biostatistics Department at ISGlobal (generic institutional email: info@isglobal.org; sergi.sanz@isgloabal.org).

**Funding:** The Agencia Española de Cooperación Internacional para el Desarrollo (AECID) support the implementation of the work, through a Development Agreement (grant number 14-CO1-558) (MJP, MRC, RS, WGR, CR, JJPR, LO, MC, NMC, DL, FT, JG). MJP and JG research is supported by Departament d'Universitats, Recerca i Societat de la Informació (grant number AGAUR 2017SGR00924), the Instituto de Salud Carlos III (ES) RICET Network for Cooperative Research in Tropical Diseases (grant number ISCIII RD12/0018/0010 FEDER) and Centro de Investigación Biotecnológica en Red de Enfermedades Infecciosas (CIBERINFEC)- (grant number CB21/13/00112). CIBERINFEC is co-funded with FEDER funds. MJP research is also supported by the Conselleria de Sanitat Universal i Salut Pública (grant number PERIS 2016-2010 SLT008/18/00132). The funders had no role in study design, data collection and analysis, decision to publish, or preparation of the manuscript.

**Competing interests:** The authors have declared that no competing interests exist.

## Conclusions

After being recognized by the Chagas National Programme as a healthcare model aligned with national laws and priorities, the Bolivian platform of Chagas as an innovation, includes attributes that they have made it possible to expand the strategy at the national level and could also be adapted in other countries.

### Author summary

The Bolivian Chagas Platform was born in 2009 to promote comprehensive care for Chagas disease (CD), a neglected tropical disease that affects more than a million people in Bolivia. A two-phase strategy was designed to introduce protocol-based healthcare in Bolivia through prevention, case-management, healthcare professionals training, and community activities. From an initial seven centers in the vertical phase (Chagas Platform centers), 52 healthcare primary healthcare centers adopted CD protocolized care in a second phase (Chagas Healthcare Network) through decentralized, horizontal scaling-up. 181,397 people at risk of having *T. cruzi* infection were tested (15%), 57,871 (31.9%) tested positive, and 18,582 (32.1%) were treated. Sixty-seven percent of healthcare workers were trained. Adequate domestic financial and human resources were ensured at the end of the scaling-up. Translational research and training activities improved evidence-based decision-making in clinical management. The Bolivian Chagas Platform as innovation, included attributes that enabled scaling-up at national and international level.

## 1. Introduction

Chagas disease (CD) is a neglected tropical disease, endemic and highly prevalent in Latin America, caused by *Trypanosoma cruzi*.[1] Transmission of CD can take place via insect vectors or other routes (mother-to-child, blood-transfusion and organ transplant). [1] In recent decades, migratory flows made CD a globalized health problem with ~7 million people affected and 25 million at risk of infection. [2] The estimated annual global economic burden of CD in healthcare costs alone is over $600 million. [3]

Although CD was described more than a century ago, there are still many challenges in its management, from prevention to early diagnosis and treatment. Symptoms in the acute stage of infection are usually non-specific or nonexistent, complicating diagnosis and timely treatment. Without treatment, the infection becomes chronic and leads to 40% of people developing life-threatening ailments in the heart and/or digestive tract organs. [4]

Bolivia has the highest prevalence of CD in the world (6·1%).[1] Despite CD being a priority for the Bolivian Ministry of Health (MoH), [5] as in other countries, there is no regulation for implementing comprehensive care for people at risk of having *T. cruzi* infection. [6] Due to the scarcity of resources, in the 90's and early 2000's, prevention strategies focused on vector control and mother-to-child transmission. By 2009, there was no comprehensive strategy for management of *T. cruzi* infection in adults. In fact, only a small percentage of people with Chagas receive care and treatment worldwide. [7]

With the aim of reinforcing the Chagas National Programme (ChNP) of the Bolivian MoH and improving CD control, we developed the Bolivian Chagas Platform in 2009. [8] The first stage of the project (2009–2014) implemented a vertical pilot program to introduce protocol-based healthcare using a top–down strategy in coordination with local health authorities. [8]

The second phase (2015–2018) aimed at scaling up this vertical model to primary healthcare centers with a comprehensive and realistic horizontal approach, simplifying processes, and building the Chagas Healthcare Network (Fig 1).

The Chagas Platform was developed by the Barcelona Institute for Global Health (ISGlobal) and CEADES Foundation (CEADES), in collaboration with Bolivian Ministry of Health, Bolivian Chagas National Programme and local institutions (Mayor de San Simon University-Cochabamba, Juan Misael Saracho University- Tarija and civil society associations of patients at risk of having CD).

The aim of this manuscript is to evaluate the scaling-up process we implemented and to report the main results from this eight-year project.

## 2. Methods

### 2.1. Ethics statement

The specific intervention did not require ethic committee approval. Nevertheless, patients entered in the Chagas Platform and Chagas Network system signed and informed consent before starting treatment with benznidazole or nifurtimox. The clinical management protocol and informed consent documents were reviewed and approved by the Ethical Committees of Fundación CEADES (Cochabamba, Bolivia).

### 2.2. Intervention area

The project intervention area were three CD endemic departments (Cochabamba, Tarija, and Chuquisaca) in Bolivia.

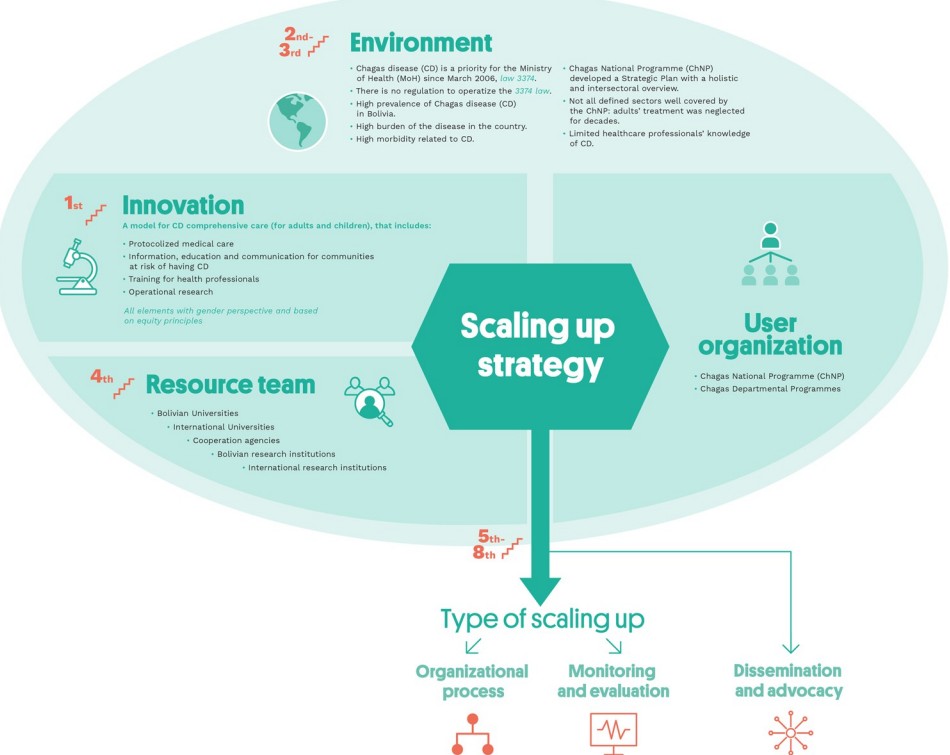

**Fig 1. Elements of Chagas Platform at scaling up strategy.** Credits:ISGlobal.

In the first phase, seven municipalities (three in urban and four in rural areas) were selected to implement reference centers (Bolivian Chagas Platform centers). For scaling up (second phase) the intervention area was expanded based on a collaborative selection with the Chagas directors of the above-mentioned departments (decentralized process supported by 2010/031 Bolivian law), [5] covering two-thirds of three CD endemic departments.

### 2.3. Conceptual framework for scaling up

Scaling up has been defined as "deliberate effort to increase the impact of successfully tested health innovations so as to benefit more people and to foster policy and Programme development on a lasting basis" [9]

Based on this definition, the World Health Organization (WHO) created ExpandNet, [10] a global network that seeks to promote equitable access to quality care by facilitating scaling-up processes. ExpandNet has developed an evaluation and review strategy for those interventions already implemented or in progress.

The nine steps proposed by ExpandNet to ensure a systematic approach to institutionalizing and expanding innovative experiences are listed in Box 1. [11]

### 2.4. Implementation of strategies

The first phase of the project was a vertical strategy (the Chagas Platform). [8,12] In this first phase, the goal was to build a comprehensive care model for patients with Chagas in collaboration with Departmental Health Authorities belonging to the Ministry of Health. By its implementation, population use and the increase of healthcare demands were tested to ensure not only the effectiveness of the action, but the future acceptance by health authorities and also by health professionals and community.

---

## Box 1. The nine steps

Step 1. Planning actions to increase the scalability of the innovation

Step 2. Increasing the capacity of the user organization to implement scaling up

Step 3. Assessing the environment and planning actions to increase the potential for scaling-up success

Step 4. Increasing the capacity of the resource team to support scaling up

Step 5. Making strategic choices to support vertical scaling up (institutionalization)

Step 6. Making strategic choices to support horizontal scaling up (expansion/ replication)

Step 7. Determining the role of diversification

Step 8. Planning actions to address spontaneous scaling up

Step 9. Finalizing the scaling-up strategy and identifying next steps

**Source:** Nine steps for developing a scaling-up strategy. Department of Reproductive Health and Research. WHO Library Cataloguing-in-Publication Data.2010

---

The primary objective of the Chagas Platform is to contribute to the control of CD, and the model designed to achieve this objective is based on four pillars:

1. Provision of care, based on clinical protocols and guidelines.

2. Training of health professionals in the management of CD

3. Creation of expertise in CD management and capacity building for research

4. Promotion of educational activities in the community

Besides providing care to their professionals, the Chagas Platform also trained professionals at the level of primary care, as key issue for the scaling-up. Although the clinical guidelines were adapted to local reality, care was taken to follow the international guidelines.

Moreover, the Chagas Platform, has collaborated since its inception with the usual preventive activities of the Chagas National Programme focused on entomological surveillance, spraying vector-infected houses, and carrying out educational actions and community engagement.

The integration of Chagas Platforms protocols into the National Health System were facilitated by the twelve-steps methodology designed by the Platform team. The twelve-steps are summarized in Fig 2.

A simplified clinical protocol for comprehensive care was considered easy to replicate by local healthcare institutions in order to ensure sustainability as well as high-quality standards. The scaling up process was conducted in a gradual way in order to ensure quality of processes and monitoring. In each area the organizational process, included referral and counter-referral pathways which were locally adapted in consultation with local healthcare personnel and

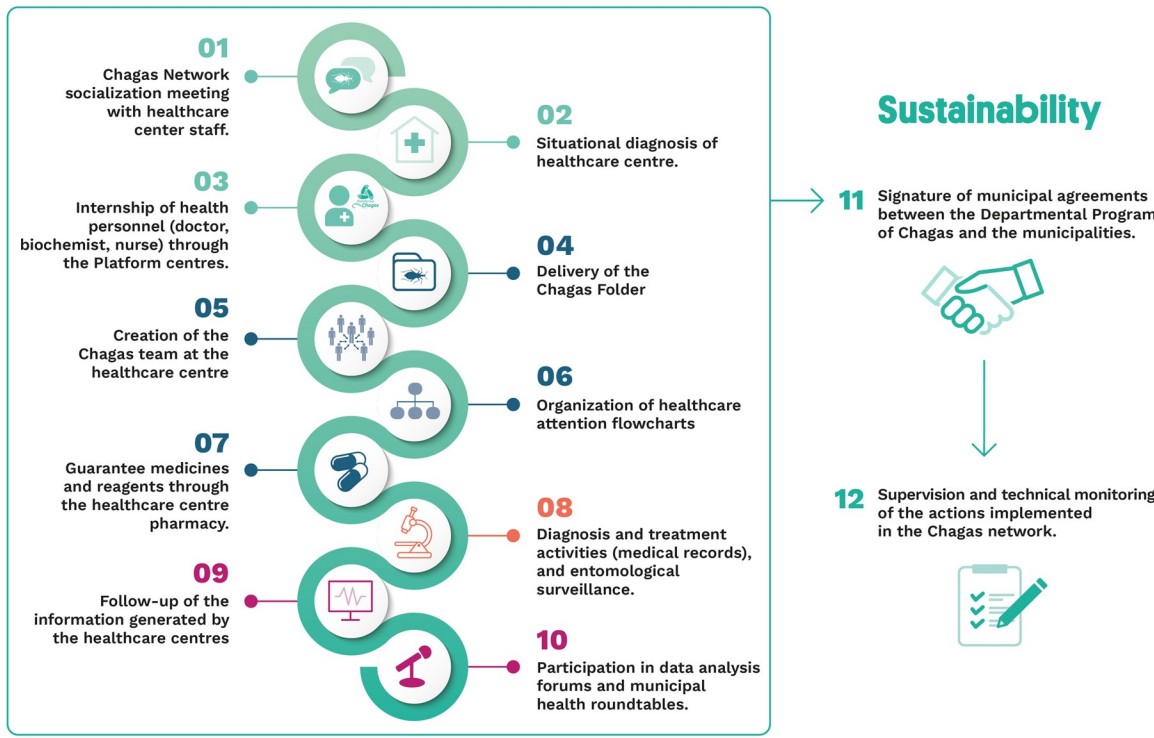

**Fig 2. The Chagas Platform Twelve Steps to scaling-up.** Credits: iStock.com/ass29.

directors. The Chagas Platform centers established during the first phase acted as technical referral centers.

Regarding the economic sustainability of actions implemented initially with the support of foreign resources, costs and resource mobilization are being including in the strategy in the second phase of the scaling up process. Facing the end of the external funding, Bolivian local institutions are currently including in their yearly budgets to support the Platform model at large scale.

## 2.5. Data collection

ISGlobal–CEADES designed and managed a database for registering Chagas Platform centers activities: clinical variables (including electrocardiographic and echocardiographic data), laboratory variables, and other tests results; entomo-epidemiological surveillance data. Specific forms (i.e., first visit, pre-treatment, treatment follow-up, post-treatment follow-up, last follow-up) were used to collect patient data in each healthcare visit. A simplified database and forms were developed for the Chagas Healthcare Network with local healthcare authorities to compile the most relevant information in a sustainable way. Reports on training, research and information, education and communication activities were registered and compiled.

## 2.6. Evaluation of scaling-up and identification of actions needed

To evaluate the Chagas Platform scaling-up, ISGlobal–CEADES followed the guideline "Nine steps for developing a scaling-up strategy" developed by ExpandNet, a global network created by the WHO (Box 1). [9–11] Details of each step analysis and evaluation of the project are included in Supporting Information (S1 Text and S1 Table) Continuous monitoring and evaluation were devised together with local authorities and implemented to assess processes (clinical management, diagnosis techniques, research protocols, educational activities) and outcomes based on local and international quality standards.

## 3. Results

### 3.1. Coverage and CD management in Chagas Healthcare Network

In terms of Health Coverage, from seven initial centers implemented in the vertical phase, 52 healthcare centers belonging to the primary healthcare national system network adapted Chagas disease protocolized care through the scaling up strategy proposed in three out the four highly endemic departments (Cochabamba, Chuquisaca and Tarija) (Fig 2)

By the end of the second stage of the project (December 31th 2018), 181,397 people at risk of having *T. cruzi* infection has been tested (15% of the intervention area population). Among them, 57,871 (31·9%) had a confirmed to have *T. cruzi* infection of whom 18,582 (32·1% of those *T. cruzi* positive) eligible, and consenting subjects initiated treatment. A total of 14,725 completed treatment (79·2%). Treatment rates were affected by stockouts of benznidazole for 15 months due to an interruption in production; and later, for 12 months due to a higher-than expected demand.

Of the 2,284 (19%) patients who initiated treatment but did not complete it, 1,037 discontinued voluntarily and 1,247 abandoned following medical recommendation, mostly due to Adverse Drugs Reactions (ADRs) (S2 Table).

A summary of the main results by municipality and department, and by gender is expressed in Table 1. A comparison between coverage during vertical and horizontal stages is summarized in Table 2. Table 3 summarizes data on adherence and adverse events severity and incidence.

**Table 1. Chagas network health coverage (2015–2018).**

| Departmental network | Number of municipalities involved | Population | | People screened (n, % of population) | | T. cruzi infection confirmed (n, % among people screened) | | Antiparasitic treatment administered (n, % among T.cruzi positive people) | |
|---|---|---|---|---|---|---|---|---|---|
| | | F* | M** | F | M | F | M | F | M |
| Cochabamba | 12 | 312,631 | 307,668 | 64,421 (20·6%) | 39,082 (12·7%) | 12,338 (19·1%) | 4,113 (10·5%) | 459 (3·7%) | 338 (8·2%) |
| Chuquisaca | 12 | 198,942 | 192,445 | 12,612 (6·3%) | 8,628 (4·5%) | 3,705 (29·4%) | 1,973 (22·9%) | 557 (15%) | 372 (18·8%) |
| Tarija (Chaco) | 3 | 85,193 | 91,342 | 3,102 (3·6%) | 2,104 (2·3%) | 887 (28·6%) | 551 (26·2%) | 98 (11%) | 96 (17·4%) |
| **Total** | **27** | **596,766** | **591,455** | **80,135 (13·4%)** | **49,814 (8·4%)** | **16,930 (21·1%)** | **6,637 (13·3%)** | **1,114 (6·6%)** | **806 (12·1%)** |

* F: female

**M: male; n: absolute number.

## 3.2. Training, educational interventions, research and policy: Interaction with stakeholders

In terms of training, 321 out of the 479 healthcare workers in the intervention area (67%) were trained through both theory and practical stages at Chagas Platform reference centers.

The Chagas Platform technical team reviewed the Bolivian National Guidelines for CD (under development) together with the National Chagas Programme authorities to align prevention, diagnosis and treatment strategies. We implemented a simplified version of the guidelines (validated by municipal and department authorities), consisting of a summary of practical indications easy to manage by healthcare workers. This implementation modified and consolidated diagnosis and evidence-based treatment practices. Diagnosis tests were homogenized and changed from indirect hemagglutination and low sensitivity immunochromatographic test to a wider use of enzyme-linked immunosorbent assays, with higher sensitivity and specificity.

In terms of prevention, scaling-up of the entomological surveillance with community engagement achieved higher rates of house infestation reports than with house-by-house vector control by technicians, leading to more effective and sustainable spraying [13].

Regarding governance reinforcement, besides development of technical documents such as protocols and guidelines, a sustainability plan was created with local authorities. A key milestone was signing an agreement with municipal authorities to maintain long-term comprehensive care for CD. Additionally, local resources to financially support this were attributed in 78% of Cochabamba municipalities, 87% of Chuquisaca municipalities, and 50% of Tarija municipalities.

**Table 2. Chagas Platform and Chagas Healthcare Network coverage (2010–2018).**

| | People screened (total population) | T. cruzi infection confirmed (n, % among people screened) | Antiparasitic treatment administered (n, % among T. cruzi positive people) | Antiparasitic treatment finished (n, % among people that starts treatment) |
|---|---|---|---|---|
| **Chagas Healthcare Network** | 129,949 | 23,567 (18·1%) | 1,920 (8·1%) | 1,370 (71·3%) |
| **Chagas Platform centers** | 51,448 | 34,304 (66·7%) | 16,662 (48·6%) | 13,355 (80·1%) |
| **Total** | **181,397 (15%)** | **57.871 (31·9%)** | **18,582 (32·1%)** | **14,725 (79·2%)** |

n: absolute number.

**Table 3. Antiparasitic treatment outcomes and benznidazol/nifurtimox tolerability description, in Chagas Platforms centers and Chagas network (2010-06/2018).**

| | | Antiparasitic treatment administered (n) | Antiparasitic treatment finished (n) | ADRs* reported (n) | | | Treatment stopped | | | Treatment abandoned by patient |
|---|---|---|---|---|---|---|---|---|---|---|
| | | | | Mild | Moderate | Severe | ADRs | Pregnancy | Others | |
| Chagas network | F** | 1,114 | 820 | 217 | 73 | 27 | 157 | 2 | 9 | 126 |
| | M*** | 806 | 550 | 123 | 38 | 20 | 89 | - | 8 | 159 |
| Chagas Platform centers | F | 6,163 | 5,034 | 2,639 | 898 | 181 | 604 | 25 | 47 | 452 |
| | M | 3,928 | 3,323 | 1,049 | 281 | 125 | 269 | - | 37 | 300 |
| Total (n, %) | | 12,011 | 9,727 (80·9%) | 4,028 (33·5%)**** | 1,290 (10·7%)**** | 353 (2·9%)**** | 1,119 (9·3%)**** | 27 (0·2%)**** | 101 (0·8%)**** | 1,037 (8·6%)**** |

*ADRs: Adverse Drug Reactions

** F: female

***M: male

****% related to number of treatments started; n: absolute number.

Translational research was carried out: 28 research protocols were presented to a scientific committee during the eight years of intervention, 24 of them in the second stage (the last four years). This project also led to several published research articles, [8,12–20] and others still under development.

Research activities indirectly impacted governance and regulatory agencies, promoting and improving regulation for carrying out clinical trials in Bolivia. Having well informed patients and well-trained teams in research centers identified further CD research possibilities, increasing income to maintain Platform center structures and expand research expertise in the three intervention departments. [16, 20–22]

Dissemination was carried out in national, regional and international meetings, presenting the model and main results of the Chagas Platform comprehensive care strategy, research, training and community activities. Additionally, two internationals meetings on CD were organized in 2013 and 2016 in Bolivia.

Activities in schools and the community (health fairs, seminars, workshops) were implemented and continuously carried out (Table 4), resulting in an increased demand for diagnosis and treatment compared with MoH data prior to the intervention.

## 4. Discussion

We evaluated the scaling up of a comprehensive strategy to address CD in Bolivia and found that it was successful in terms of healthcare coverage, quality of the process, adaptability to different epidemiological scenarios and mid- and long-term sustainability.

In terms of healthcare coverage, it is important to highlight that, during the four years of scaling-up, at-risk people managed in the intervention areas increased by 160% with the

**Table 4. Community based activities performed in the context of Chagas network (2015–2018).**

| Departmental Network | N° community activities | N° of persons informed | | |
|---|---|---|---|---|
| | | In Quechua | In Spanish | TOTAL |
| Cochabamba | 63 | 2,100 | 3,605 | 5,705 |
| Chuquisaca | 36 | 1,446 | 1,139 | 2,585 |
| Tarija (Chaco) | 36 | 0 | 1,002 | 1,002 |
| INTERVENTION AREA (Bolivia) | 135 | 3,546 | 5,746 | 9,292 |

comprehensive approach proposed by the Chagas Platform, compared to the rates achieved in the eight years of vertical approach. Data showed that Chagas Platform approach, based on specialized centers, is more efficient (higher rate of positive cases among people tested). Nevertheless, in terms of sustainability and access it is less effective, given that Chagas Platform centers are only seven in the country, are located in specific places and depends mainly on external funding. Thus, data coming from Chagas Platform patient's prevalence is under a selection bias. People attending Chagas Platform centers were in most cases people that knew about their specific risk or even they have a previous diagnosis on Chagas, and that aimed to be diagnosed and treated. In some cases they were relatives of previously diagnosed people, or positive cases derived from other healthcare centers to be treated. In terms of access to treatment, absolute number of patients treated in these seven centers, belonging to Chagas Platform, is lower than in the Chagas Healthcare Network, and it does not mean a problem in terms of drugs stock for the National Health System as drug provider. On the other hand, the most of the people attending Chagas Platform centers were young people without treatment contraindications.

On the contrary, people attending primary healthcare centers belonging to Chagas Healthcare Network are people that usually attend the healthcare center for any reason, and *T.cruzi* screening is included in their regular management, but in a protocolized way. In this case, prevalence observed is closer to reality than the one observed in the Chagas Platform centers. Regarding specific treatment, the profile of patients attending primary healthcare centers was slightly different from those attending the Chagas Platform, having a percentage of them contraindications to start etiologic treatment due to age and/or organ involvement.

One of the key actions undertaken from the beginning of the project in 2009 was working with local stakeholders to progressively transfer responsibility. Decentralization laws (2010/031 Bolivian law) [23] allowed a coordination team to work together with department and municipal healthcare directors that were also field executors and practitioners.

During this eight-year project, constant changes in the political environment led to high staff turnover, resulting in the coordination team devoting significant time and effort to educating and training ChNP newcomers. Although dialogue with local authorities was complicated when addressing intersectoral issues, that included educational, housing and environmental sectors interacting with health sector, the effort to promote and build a comprehensive response led to more robust results when addressing CD. In Bolivia, there is scope to enhance constructive dialogue between civil society, politicians and decision-makers, but community pressure can influence policy: accurately informing civil society and providing communication materials have a positive impact in aligning community needs, health policy and innovative solutions.

We developed didactic material using one of the main indigenous languages, Quechua, providing closer links with the community to understand its points of view and address its needs. There is little published on using indigenous languages to improve healthcare, usually focused on social approaches to entomological prevention. [24,25] Collaborative strategies with the community should be implemented in order to strength information, education and communication activities. [25]

One of the main achievements of the Chagas Network scaling-up process was ensuring adequate domestic financial and human resources for comprehensive CD care in the intervention areas. This was the result of considerable effort advocating for comprehensive CD care activities to be included in annual healthcare municipal budgets.

Another achievement was the inclusion of CD indicators in municipal and department data analysis meetings, promoting the evaluation of the program and continuous discussion of CD by health policymakers at municipal and department levels, helping raise the profile of the disease.

Training is a key element in institutionalization and scaling-up of strategies, and should cover training and continuous evaluation of healthcare personnel, and training reinforcement in public health, data analysis, planification and evaluation. [19]

Leadership of women was promoted as a transversal approach throughout the two phases of the intervention, mainly in activities related with community engagement (at community level) and regarding healthcare personnel specialized training and leading tasks. An in-depth analysis of this will be described in a separate manuscript.

While scaling up, the Chagas Platform model was adapted easily to each scenario with the twelve-step methodology, starting with a situational analysis of each area to maximize resources and to establish referral and counter-referral pathways. There was variability in the pace of implementation in each area and in the issues that needed reinforcement and closer supervision. In all cases, participation of all stakeholders at different levels when required, resulting in a strong model.

Regarding diagnosis, one of the main achievements of the strategy was to homogenize the diagnostic test with high sensitivity (Se) and specificity (Sp) ELISA techniques embed in National Health System regular practices. Besides, and in favor of increasing access for people far from healthcare facilities, research on higher Se and Sp rapid diagnosis tests was carried out during these years to potentially implementing them in the future. [16,21]

Translation of such research results, that promotes changes in healthcare practices and approaches, is difficult to quantify in terms of impact in the short-term, but it has been a key factor to develop and implement the model proposed with success. This approach implies strengthening research capabilities in local teams, having as a result a stronger healthcare professional, and a country with the capability to generate knowledge to fight against public health threats. In this regard, collaborations at international level such the improvement of decentralized laboratory network [26] and the possibility to implement clinical trials was key. [15,18]

Dissemination of preliminary results led to an increased demand to diagnose and treat children and people with organ damage secondary to CD. Healthcare quality, training activities and interconnection of different healthcare system levels facilitated this, with very good acceptance from the community and user organizations.

In recent years, other strategies have been implemented in Latin America to improve CD control but only on a pilot level. In a region of Argentina, a customized program for early diagnosis and treatment for CD showed patients could be diagnosed in primary healthcare by providing support with capacity building. [27] In a region of Colombia, a collaborative pilot project was carried out to increase diagnosis and access to treatment for CD, in order to validate a patient centered roadmap, but with limited real impact on the population's health. [28] All these strategies, together with specific actions are catalyzed and shared at international level with the leadership of Chagas Global Coalition, in order to promote synergies among stakeholders. [29]

Along the implementation of the Chagas Platform and the subsequently scaling-up process, several limitations were raised, and then, confirmed when the evaluation was performed at the model (S1 Table). In terms of technical sustainability external technical advice is still needed in the Chagas Healthcare Network at primary healthcare level. Another relevant example is the current surveillance system: the Bolivian national health information system has not been harmonized with indicators selected by local healthcare personnel. Monitoring and supervision tools were developed by local healthcare directors, but not validated on a national healthcare level.

In conclusion, the Chagas Platform Network was a successful model for scaling up diagnosis and treatment of CD in Bolivia and empowering local healthcare centers and personnel. We used the 9th step of the ExpandNet guideline to highlight recommended actions and

prioritize them for future work (S1 Table). Translational research activities and training of healthcare personnel improved quality of care and increased evidence-based decision-making in CD clinical management. Nevertheless, several challenges remain to be addressed to ensure medium- and long-term sustainability. In countries where CD is endemic, quality of diagnosis and treatment should be improved, and communication and dissemination activities should be reinforced in order to support institutions, healthcare practitioners and decision-makers.

## Supporting information

**S1 Text. Supplementary results: Identification of the actions needed for scaling-up following steps 1 to 8.**
(DOCX)

**S1 Table. Step 9.** Finalizing the scaling-up strategy and identifying next steps: Recommended actions in the scaling-up steps.
(DOCX)

**S2 Table. Antiparasitic treatment and ADRs in Chagas Platforms centers and Chagas network.**
(DOCX)

## Acknowledgments

The authors thank Bolivian Chagas patient's associations, the Chagas Global Coalition and personnel belonging to Bolivian Chagas Departmental Service at municipal and departmental level for their support on implementing the Chagas Platform and the Chagas Healthcare network. We would also specially thanks Dr. Kate Ralston for her critical reading of the article and Carme Subirà for her contribution of figures editing.

Membership of Chagas Platform and Chagas Healthcare Network working group: **ISGlobal-CEADES Platform for Comprehensive Care of Patients with Chagas Disease, Bolivia (Chagas Platform**): Cochabamba: Gimena Rojas Delgadillo, Helmut Ramón Magne Anzoleaga, Wilma Chambi, Maria Yurly Escobar Caballero, Seyla Gamboa, Rose-Marie Arze Tapia, Karen Toledo, Jimena Ramos Morales, Lilian Pinto, Goretti Rojas, Roxana Challapa Quechover, Mery Arteaga, Mery Arteaga, Dunia Torrico Terceros; Punata: Jareth Sanchez, Albaro Pardo, Daniela Camacho, Eliana Claros; Sacaba: Alejandro Camacho, Lucia Barra, Elia Chavez, Gloria Sandy; Villa Tunari: Lizeth Rojas Panozo, Eliana Ugarte, Josue Condori, Silvia Vigabriel, Esther Sejas, Fernando Estrada, Cinthia Martinez; Nueva Gante: Gisela Vidal, Israel Vidal, Cecilia Huanca; Tarija: Alejandro Palacios, Yalu Gallardo, Isabel Gonzales, Laura Vallejos, Rudy Vasco, Patricia Martinez, Eduardo Romero, Maria Eugenia Flores; Sucre: Roger Arteaga Mejía, Marisabel Chavarría Arroyo, Paola Paco Mamani, Edwin Vasquez Cruz, Ana Daza Sanchez, Liliana Vidal Flores, Karen Portillo Ferrufino. CEADES Foundation-Bolivia: Felix Marca Mamani. Hospital Clínic-ISGlobal: Mònica Solanes Valverde, Elizabeth Posada Diago.

**Chagas Healthcare Network (Bolivian National Health System):** Head Chagas National Programme: Enzo Gamarra; **Cochabamba:** Chagas departmental Programme Cochabamba: Rossemary Grageda, Rene Fernandez; Chagas Healthcare Network at Cochabamba: Ricardo Cespedes Sanabria, Efrain Valencia, Walter Tarifa Rodriguez, Bernardina Herrera Escalera, Ondina Castellon, Geovanna Quispe, Fidel Fernandez, Gerald Marcus, Paola Dominguez Pozo, Carmen Romero, Lizbeth Lara Montaño, Sergio Sandoval Zurita, Angelica Mendoza Untoja, Jorge Armando Tordoya, Gabriel Sambrana, Fernando Aguilar, Griselda Zurita Sejas, Geovana Garcia Ignacio, Maria Maldonado, Josue Condori, Richard Roby Rojas Moya, Homer Maldonado, Rolando Meneses Calatayud, Victor Villarroel, Franz Gery Rojas Sanjines,

 

Leydi Vanesa Fernandez, Carla Lorema Colque Torrez, Cinthia Rojas, Lenny Romero Orellana, Jorge Herrada, Maria Litzi Vargas Arias, Wilfredo Montaño, Danitza Pereira Ascarraga, Daniel Hervas Choque, Cuper Maldonado Panozo. **Chuquisaca:** Chagas departmental Programme Chuquisaca: Roberto Loredo, Johnny Camacho, Karina Egüez. Chagas Healthcare Network at Chuquisaca: Ruth G. Carranza Calderón, Abel Bustillos Fortún, Maribel Gonzales Miranda, Ernesto Pinto Alegre, Vladimir Alvarez, Gregorio Nuñez Gonzales, Leticia Balderrama López, Primitivo Oblitas, Juan Pablo Vedia Zárate, Zulma Lucas, Jenny Gutierrez, Williams Gardeazabal, Maria Luisa Rocabado, Oscar Rejas Cáceres, Isaac Caballero, Fidelia Durán Parina, Noelia Ramirez Salazar, Nelson Huayllas Alejo, Ruben Ortiz Picón, River Rister Zardán, José Enrrique Melgarejo, Patricia Maldonado, Adhemar Danny Alvarez López, Cristhian Larrazábal Benitez, Jhonny Balcera Paco, Ybeth Crespo Maldonado, Betty Flores Llanes, Hector Meras Durán, Carla Palenque, Lusman Saavedra Vargas, Luis Alberto Muñoz, Juan Mendoza Coa; **Tarija**, Chagas departmental Programme Tarija: Eduardo Rueda Guerrero, Sebastia Nelly Aguado Aparicio. Chagas Healthcare Network at Chaco Tarija: Arsenio Erasmo Tapia Sanchez, Enueli Aguado Aparicio, Margarita Miranda Lopez, Silvio Bruno Morales Alvis, Norma Edith Moscoso, Cilene Valdez Velasco, Victor Hugo Huarachi, Bertha Mendoza Prado, Esther Karina Albino Condori.

## Author Contributions

**Conceptualization:** Maria-Jesus Pinazo, Mirko Rojas-Cortez, Jimy-Jose Pinto Rocha, Lourdes Ortiz, Daniel Lozano, Faustino Torrico, Joaquim Gascon.

**Data curation:** Maria-Jesus Pinazo, Mario Castellon, Daniel Lozano.

**Formal analysis:** Maria-Jesus Pinazo, Jimy-Jose Pinto Rocha, Lourdes Ortiz, Mario Castellon, Daniel Lozano, Faustino Torrico, Joaquim Gascon.

**Funding acquisition:** Maria-Jesus Pinazo, Faustino Torrico, Joaquim Gascon.

**Investigation:** Maria-Jesus Pinazo, Mirko Rojas-Cortez, Ruth Saravia, Wilson Garcia-Ruiloba, Carlos Ramos, Jimy-Jose Pinto Rocha, Lourdes Ortiz, Mario Castellon, Nilce Mendoza-Claure, Daniel Lozano, Faustino Torrico, Joaquim Gascon.

**Methodology:** Maria-Jesus Pinazo, Mirko Rojas-Cortez, Daniel Lozano, Faustino Torrico, Joaquim Gascon.

**Project administration:** Ruth Saravia, Wilson Garcia-Ruiloba, Carlos Ramos, Jimy-Jose Pinto Rocha, Daniel Lozano, Faustino Torrico, Joaquim Gascon.

**Software:** Mario Castellon.

**Supervision:** Maria-Jesus Pinazo, Mirko Rojas-Cortez, Ruth Saravia, Wilson Garcia-Ruiloba, Carlos Ramos, Jimy-Jose Pinto Rocha, Lourdes Ortiz, Nilce Mendoza-Claure, Daniel Lozano, Faustino Torrico, Joaquim Gascon.

**Validation:** Maria-Jesus Pinazo, Mirko Rojas-Cortez, Mario Castellon, Nilce Mendoza-Claure, Faustino Torrico, Joaquim Gascon.

**Visualization:** Maria-Jesus Pinazo, Mario Castellon.

**Writing – original draft:** Maria-Jesus Pinazo, Joaquim Gascon.

**Writing – review & editing:** Maria-Jesus Pinazo, Mirko Rojas-Cortez, Ruth Saravia, Wilson Garcia-Ruiloba, Carlos Ramos, Jimy-Jose Pinto Rocha, Lourdes Ortiz, Nilce Mendoza-Claure, Daniel Lozano, Faustino Torrico, Joaquim Gascon.

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
