## [Decision Letter · Decision Letter 0]

8 Nov 2021

Dear Dr. PINAZO,

Thank you very much for submitting your manuscript "Results and evaluation of the expansion of a model of comprehensive care for Chagas disease within the National Health System: the Bolivian Chagas network" for consideration at PLOS Neglected Tropical Diseases. As with all papers reviewed by the journal, your manuscript was reviewed by members of the editorial board and by several independent reviewers. The reviewers appreciated the attention to an important topic. Based on the reviews, we are likely to accept this manuscript for publication, providing that you modify the manuscript according to the review recommendations. 

Sincerely,

Natalie Bowman, MD

Associate Editor

Joseph Vinetz

Deputy Editor

Reviewer's Responses to Questions

**Key Review Criteria Required for Acceptance?**

**Methods**

-Are the objectives of the study clearly articulated with a clear testable hypothesis stated?

-Is the study design appropriate to address the stated objectives?

-Is the population clearly described and appropriate for the hypothesis being tested?

-Is the sample size sufficient to ensure adequate power to address the hypothesis being tested?

-Were correct statistical analysis used to support conclusions?

-Are there concerns about ethical or regulatory requirements being met?

Reviewer #1: -Are the objectives of the study clearly articulated with a clear testable hypothesis stated? Yes

-Is the study design appropriate to address the stated objectives? Yes 

-Is the population clearly described and appropriate for the hypothesis being tested? Yes

-Is the sample size sufficient to ensure adequate power to address the hypothesis being tested? Yes

-Were correct statistical analysis used to support conclusions? Yes

-Are there concerns about ethical or regulatory requirements being met? Yes

Reviewer #2: -Are the objectives of the study clearly articulated with a clear testable hypothesis stated? YES

-Is the study design appropriate to address the stated objectives? YES

-Is the population clearly described and appropriate for the hypothesis being tested? YES

-Is the sample size sufficient to ensure adequate power to address the hypothesis being tested? YES

-Were correct statistical analysis used to support conclusions? NON APPLICABLE 

-Are there concerns about ethical or regulatory requirements being met? YES

**Results**

-Does the analysis presented match the analysis plan?

-Are the results clearly and completely presented?

-Are the figures (Tables, Images) of sufficient quality for clarity?

Reviewer #1: -Does the analysis presented match the analysis plan? Yes

-Are the results clearly and completely presented? Yes

-Are the figures (Tables, Images) of sufficient quality for clarity? Yes

Reviewer #2: -Does the analysis presented match the analysis plan? YES

-Are the results clearly and completely presented? YES

-Are the figures (Tables, Images) of sufficient quality for clarity? YES

**Conclusions**

-Are the conclusions supported by the data presented?

-Are the limitations of analysis clearly described?

-Do the authors discuss how these data can be helpful to advance our understanding of the topic under study?

-Is public health relevance addressed?

Reviewer #1: -Are the conclusions supported by the data presented? Yes

-Are the limitations of analysis clearly described?

-Do the authors discuss how these data can be helpful to advance our understanding of the topic under study? Yes

-Is public health relevance addressed? Yes

Reviewer #2: -Are the conclusions supported by the data presented? YES

-Are the limitations of analysis clearly described? YES 

-Do the authors discuss how these data can be helpful to advance our understanding of the topic under study? YES

-Is public health relevance addressed? YES

**Editorial and Data Presentation Modifications?**

Reviewer #1: Minor corrections:

Line 184 – in to - correct to into

Line 247 Table 3 – please explain what are the categories “Chagas network” and “Chagas Platform centers” . The title proposes “Chagas network” and in Table 3 there is a division between “Chagas network” and “Chagas Platform”. Please define better these categories.

Line 295 – please correct Depatmental Network to Department Network

Line 311 – what is ChNP ? Chagas network platform? 

Line 312 – Please explain this sentence “Although dialogue with local authorities was complicated when addressing intersectoral issues, it led to more robust results when addressing CD.”

Reviewer #2: The manuscript titled "Results and evaluation of the expansion of a model of comprehensive care for Chagas disease within the National Health System: the Bolivian Chagas network" is a great description and evaluation about an impressive health primary care model built to achieved health access for people affected by Chagas disease. I think that for the main purpose of this manuscript do not need any correction , but after read it I have some question of personal interested if authors are willing to answer or discussed :

1) besides serologies and treatment, do you have any data about EKG or cardiologic clinical evaluation? Any support/ contacted to transfer patients whose need second or third level health assistance?

2) Paragraph (lines 301-304) “ In terms of healthcare coverage, it is important to highlight that, during the four years of scaling-up, at-risk people managed in the intervention areas increased by 160% with the comprehensive approach proposed by the Chagas Platform, compared to the rates achieved in the eight years of vertical approach”

I understand that much much more people of risk were tested but in table 3, the prevalence of seropositives in the initial Platform were 66,7% while in the scale up network was 18,1 % , both prevalence are really high, but the first one extremely high ( at least one of two people tested were positive!) Why do you think is this difference prevalence between the platform and the network?

3) In the same table 3 there is a high difference in people treated between platform ( 48,6%) and platform ( 8,1%), could be this associated to clinical trials? 

4) Table 4, 1000 people abandoned treatment, may be many reasons but could be possible to know why? So if there is any cultural o social barrier the health care system could do a better approach .

5) Paragraph (lines 267-271) “In terms of prevention, scaling-up of the entomological surveillance with community engagement achieved higher rates of house infestation reports than with house-by-house vector control by technicians, leading to more effective and sustainable spraying (Rojas Cortez M, et al..)”. It is possible to know if have find infected houses? If yes, when (which year) was the last one find?

**Summary and General Comments**

Reviewer #1: This is a very interesting paper reporting on a unique experience related to the organization of healthcare for patients with Chagas in Bolivia, a country with the highest prevalence of Chagas in the world. The collaborative strategy set up as the Bolivia Chagas Network is a extremely creative solution that was developed in 3 departments and the results clearly show a very positive evaluation concerning the number of people included in the healthcare strategy, the number of affected persons treated and the success of completing treatment in about 75% of the patients. Intensive training of the health care local workforce for the management of Chagas disease care was also an important result. Adequate methodology and discussion are presented. To my knowledge this is the first and widest approach to improve Chagas disease care integrating so many primary units, beyond other pilot experiences. The paper should be published after some minor corrections, listed above.

Reviewer #2: (No Response)

PLOS authors have the option to publish the peer review history of their article (what does this mean?). If published, this will include your full peer review and any attached files.

Reviewer #1: No

Reviewer #2: No

Figure Files:

Data Requirements:

Reproducibility:

References

---

## [Editor Report · Decision Letter 1]

7 Dec 2021

Dear Dr. PINAZO,

We are pleased to inform you that your manuscript 'Results and evaluation of the expansion of a model of comprehensive care for Chagas disease within the National Health System: the Bolivian Chagas network' has been provisionally accepted for publication in PLOS Neglected Tropical Diseases.

Best regards,

Natalie Bowman, MD

Associate Editor

Joseph Vinetz

Deputy Editor

This was a nice article describing an important public health intervention to increase diagnosis and treatment of Chagas disease at the population level.

Please make sure to review for grammar and punctuation in final proofs. For example (not inclusive):

- for numbers such as 4500, use comma (4,500) rather than period (4.500), as were used in some tables. Also sometimes for 4-digit numbers, a period is used and others not, be consistent.

- please do not use contractions ("doesn't" on line 313, for example)

- I think in lines 303-307, confirm that efficient and effective are used correctly - I would interpret a strategy that diagnoses a higher prevalence to be efficient vs a higher number of people (effective).

- Figure 2's title is in Spanish

---

## [Editor Report · Acceptance letter]

26 Jan 2022

Dear Dr. PINAZO,

We are delighted to inform you that your manuscript, "Results and evaluation of the expansion of a model of comprehensive care for Chagas disease within the National Health System: the Bolivian Chagas network," has been formally accepted for publication in PLOS Neglected Tropical Diseases.

Best regards,

Shaden Kamhawi

co-Editor-in-Chief

Paul Brindley

co-Editor-in-Chief
